# Molecular Characterization and Pathogenicity of Chicken Parvovirus (ChPV) in Specific Pathogen-Free Chicks Infected Experimentally

**DOI:** 10.3390/pathogens9080606

**Published:** 2020-07-25

**Authors:** Luis Fabian N. Nuñez, Silvana H. Santander-Parra, David I. De la Torre, Lilian R. M. de Sá, Marcos R. Buim, Claudete S. Astolfi-Ferreira, Antonio J. Piantino Ferreira

**Affiliations:** 1Department of Pathology, School of Veterinary Medicine, University of São Paulo (USP), Av. Prof. Dr. Orlando M. Paiva, 87, São Paulo CEP 05508-270, SP, Brazil; fabiann7@yahoo.es (L.F.N.N.); silvanahsp@yahoo.com (S.H.S.-P.); daviddelatorreduque@gmail.com (D.I.D.l.T.); liliansa@usp.br (L.R.M.d.S.); csastolfi@gmail.com (C.S.A.-F.); 2Facultad de Ciencias de la Salud, Carrera de Medicina Veterinaria, Universidad de Las Américas (UDLA), Av. Jose Queri, Quito 170513, Ecuador; 3Institute for Research in Biomedicine, Central University of Ecuador, Quito CP E170201, Ecuador; 4Laboratory of Avian Diseases, Instituto Biológico, Av. Gaspar Ricardo, 1700, Bastos CEP 17690-000, SP, Brazil; marcosbuim@biologico.sp.gov.br

**Keywords:** chicken, chicken parvovirus, experimental infection, molecular characterization

## Abstract

Chicken parvovirus (ChPV) is an agent frequently associated with runting stunting syndrome (RSS). This syndrome has been reported in association with ChPV in many countries, including Brazil; however, studies characterizing the virus on a molecular level are scarce, and ChPV pathogenicity in day-old chicks remains unclear. The aim of the present work was to establish the molecular characteristics of ChPV, determine the pathogenicity of ChPV in SPF chicks and detect and quantify ChPV by qPCR in several tissues and chicks of different ages. The experimental challenge was performed at one day of age, and daily and weekly observations were performed and five birds from each experimental group (mock and infected birds) were euthanized to perform the different analysis. ChPV genome copies were detected and quantified by qPCR in gut, spleen, thymus, kidney, pancreas, proventriculus and bursa. Clinically, the infected group presented with diarrhea 24 h post-infection, which persisted until 42 days of age. The small intestine was distended, and its contents were aqueous and foamy. Enteritis and dilated crypts with cyst shapes were observed in intestinal segments. Acute pancreatitis associated with lymphocytic nodules, infiltrating lymphocytes and plasma cells between the pancreatic acinus was observed. Koch’s postulate was demonstrated and the genetic characterization of the VP1 gene showed that the Brazilian ChPV isolate belongs to the ChPV II group.

## 1. Introduction

Enteric disorders are considered the most important concern to poultry gut health due to their compromising effect on poultry growth. Enteric disorders have a multifactorial etiology and are associated with bacteria, fungi, protozoa and viruses [1,2,3,4,5,6]. Runting-stunting syndrome (RSS) is commonly reported in poultry production and is associated with several viruses, such as astrovirus, rotavirus, reovirus, coronavirus and parvovirus, and bacteria, including coccidia [7,8,9,10,11,12]. The clinical manifestation of RSS is characterized by ruffled feathers, diarrhea, cloacal pasting, apathy, depression, dwarfing, decreased weight gain and culling [13,14,15,16]. Furthermore, chicken parvovirus (ChPV) has been detected in several outbreaks of enteric diseases, principally in young animals showing signs of RSS.

ChPV is a nonenveloped DNA virus with a genome approximately 5 kb in length [17,18]. The genome encodes structural (VP1 and VP2) and nonstructural (NS) proteins [19]. ChPV was first reported by Kisary in 1984 [13]; after that, ChPV was detected and reported around the world in chickens showing signs of enteric diseases [20], including Brazil [5,10,21]. The main pathological findings in birds affected with ChPV are in the intestine, and these findings include enteritis and intestines filled with watery and frothy feces [14,22]. Experimental infections with the reference strain ABU-P1 reproduced RSS, but no microscopic alterations were found in either the intestine or other organs [22]. There are few studies about the pathogenicity of ChPV due to the difficulty of its isolation and propagation in vitro. These factors make it difficult to understand the pathogenesis of the virus. As of now, several ChPV parvovirus genomes have been compared with the reference sequence ABU P1 (Hungary) [19], including viral genomes from China [23], Korea [24] and the United States [4]; these comparisons are important for continuing the study of ChPV and for determining if all reported genotypes have the same pathogenesis mechanisms. Originally, ChPV diagnosis started with electronic microscopy [13] and was followed by antibody detection by ELISA [25]; however, molecular techniques are the most commonly used diagnostic methods for the detection of ChPV, and PCR is principally used to evaluate the NS gene, which is employed in the majority of molecular ChPV detection methods [6,26,27,28,29]. This NS gene is used for both ChPV and turkey parvovirus (TuPV) detection, and gene sequencing does not allow for discrimination between these viruses or for ChPV genotyping; thus, the use of VP1 gene sequencing, which is capable of both discrimination and genotyping, has been proposed [4,24].

This work aims to describe the in vivo pathogenicity and molecular characterization of ChPV based on the entire VP1 gene from a ChPV strain, which was isolated from embryonated eggs of chickens with RSS in Brazilian poultry flocks.

## 2. Results

### 2.1. Molecular Characterization of ChPV

The sequences obtained in this work from the VP1 gene of ChPV generated a sequence of 2061 bp, accession number MK440128.1, which corresponds to the complete gene sequence of VP1. The obtained VP1 nucleotide sequence was aligned and compared with previously published sequences of ChPV. The analysis of complete VP1 nucleotide sequences showed high similarity of nucleotides (NT) and amino acids (AA) between the analyzed sequences (Table 1). The sequence of the isolated ChPV showed 95–95.1% NT and 98–98.2% AA similarity to sequences from Korea; 91.6–95% NT and 94.9–98.2% AA similarity to sequences from the United States; 91.8% NT and 96.7% AA similarity to sequences from China and 88.2% NT and 93% AA similarity to reference sequences from Hungary. Compared to TuPV sequences, the obtained sequence showed low NT (77.9–79.7%) and AA (72.9–73.6%) similarity, as shown in Table 1. Phylogenetic analyses showed two well-defined groups, a ChPV group (100% bootstrap) and a TuPV group (100% bootstrap). The sequence of the ChPV isolate used in the present work clustered with a USA isolate (KM598414.1) (bootstrap 86%) belonging to ChPV group II; the ChPV reference sequence was grouped in a separate branch (97% of bootstrap) with the USA sequence (KM598415.1; Figure 1).

### 2.2. Experimental Infection—Clinical Signs

In the experimental infection, depression, lethargy, somnolence, ruffled feathers and diarrhea were observed 12 h post-inoculation. After 48 h of infection, the animals presented with cloacal pasting, dirty and wet feathers at the cloacal region caused by watery feces and ruffled feathers (Figure 2). Somnolence and lethargy increased, resulting in birds that moved less and had more apathy, in addition to an increase in diarrhea. The clinical signs of enteric disease, mainly diarrhea, were observed throughout the experiment (Table 2). The infected animals also showed differing characteristics of the feathers on their wings, namely, the wings were found folded in the outer direction. Ten days after infection, animals in the two groups were observed to have different sizes, with the infected animals appearing smaller and stunted compared with animals from the mock group. The mock group did not show any clinical signs and exhibited normal movement and healthy appearance, as described above.

### 2.3. Macroscopic Examination

The birds of the mock group did not show any macroscopic lesions in the organs at any age examined. The postmortem examination of the challenged birds showed distended coelomic cavity intestinal loops filled with aqueous and foamy content. The intestinal loops exhibited segmentations along the small intestine; there were dilated stretches as well as narrow ones, and the contents were liquid, containing foamy and undigested feed along the length of the intestine. The birds in all age groups presented with intestinal volvulus, in which there was rotation of the duodenal loop segment in the mesenteric axis that resembled a corkscrew (“J” appearance). The mesentery presented opacification in this segment, and there was atrophy of the pancreas. The birds showed persistence of the yolk sac. The rest of the organs did not show any macroscopic alteration (Table 2; Figure 3).

### 2.4. Histopathology Examination

The birds infected with ChPV showed preserved villous:crypt ratios and mild to moderate hyperplasia of crypts with the presence of crypt bifurcations, which were dilated, elongated and tortuous in different evaluation periods. From the 7th to 42nd day after inoculation, the crypts of Lieberkühn were lined by a squamous epithelium containing cellular debris and degenerated inflammatory cells that formed structures with a cyst shape; these structures were observed in the duodenum, jejunum and ileum (Figure 4). Table 3 shows the distribution of histopathological parameters of the digestive system in both groups. Between the 7th day and 28th day post-infection, necrosis of crypt cells was observed in all segments of the small intestine (*p* < 0.05). Furthermore, there was an increased number of mitotic cells in segments of the duodenum and an increased number of intraepithelial lymphocytes from the 7th to 42nd day (*p* < 0.05). The density of lymphocytes and plasma cells found in the lamina propria of the duodenum was significant (*p* = 0.05) from the 7th to the 21st day after inoculation, that of the jejunum was significant from the 14th to 21st day (*p* < 0.05) after infection, and that of the ileum was significant from the 21st to 35th day after infection (*p* < 0.05). The pancreas showed a loss of zymogene granules; lymphocytic nodules that contained an infiltrate of lymphocytes and plasma cells between the pancreatic acinus were also present, indicating acute pancreatitis. Lymphoplasmacytic mesenteritis was observed in the mesentery of the duodenal loop and the peripancreatic area, which is associated with pancreatitis. The birds of the mock group showed segments of the duodenum, jejunum and ileum within the standard of normal histology (Table 3; Figure 4).

### 2.5. Detection and Quantification of Chicken Parvovirus Genome Copies in the Different Tissues

The qPCR assay for the detection and quantification of ChPV showed that all analyzed animals were positive for ChPV in all collected organs from the first day post-infection. Peak titers of ChPV was observed in the intestines, for the ileum, followed by the jejunum and duodenum. However, it was very interesting to note that replication of the virus seemed to commence at 2–3 days post-infection for the ileum followed by the jejunum and then the duodenum. Interestingly, in this study, replication in the pancreas and an elevated number of genome copies were not observed. The gastrointestinal tract (duodenum, jejunum and ileum) showed the highest level of genome copies of ChPV at day fourteen post-infection, and the viral concentration decreased through day 42. The pancreas, thymus, liver, proventriculus and kidney showed basal virus genome titer from the first day until the end of the experiment at 42 days old. The spleen had the highest viral concentration at day 21, after which the concentration decreased (Table 4; Figure 5).

## 3. Discussion

In the present work, the pathogenicity, viral tissue distribution and molecular characterization of ChPV in chicks from a strain isolated in Brazil were determined with a demonstration of Koch’s postulates according to our previous description [21]. There are few complete genomes of ChPV available, and the complete genome refers to the ChPV reference strain ABU-P1, which was followed by viral genomes from Korea [24], the United States [4] and China [23]. Thus, these genomes possess the classical conformation of parvovirus and contain structural and nonstructural proteins; based on their VP1 gene sequences, three groups of ChPV can be identified [23]. The lack of ChPV VP1 gene sequences could likely be related to the difficulty in ChPV DNA isolation for whole VP1 gene sequencing; however, the majority of published works have included a molecular characterization of the NS gene sequence [15,16]. Nevertheless, genetic information based on NS gene sequences is not sufficient for discriminatory genotyping because there are few differences among strains [30]. Hence, we report the first molecular characterization of ChPV in Brazil based on complete VP1 gene sequence analyses, confirming that the ChPV used in the experimental infection belongs to the ChPV II group.

Around the world, ChPV has been reported and detected in healthy and sick chickens, but it has mainly been identified in chickens showing enteric disorders. This virus has the particularity that it is detected relatively more in chicks and young animals [15,16,17,18]. ChPV causes severe enteric problems that are characterized by the presence of diarrhea, high morbidity and mortality [5,31]. In the field, outbreaks that affected chickens showed many pathological alterations in the intestine upon postmortem examination, showing mainly distended intestines filled with aqueous feces and foamy and undigested feed. Moreover, the challenge with outbreaks in the field is the association of ChPV with other enteric viruses or bacteria, which could decrease the quality of the gut integrity [4,16,22,23,31]. Experimental infections with isolated ChPV (ABU-P1) have demonstrated that the virus causes enteric diseases, resulting mainly in chickens with diarrhea, cloacal pasting, impaired growth, runting and stunting [32]. As reported in the present investigation, the principal clinical signs present in the infected animals were also diarrhea, cloacal pasting, somnolence, apathy, ruffled feathers, impaired growth, runting and stunting. However, the ChPV strain ABU-P1 does not cause any macroscopic lesions in enteric tissues [22,32].

Interestingly, this investigation revealed the presence of intestinal volvulus characterized by the rotation of the duodenal loop segment in the mesenteric axis; thus, the duodenal loop was rolled, resembling a corkscrew and pancreatic atrophy was also observed. Lesions were previously described in commercial chicken flocks affected with RSS and reported by our own group [21]; the duodenal loop presented the same features, demonstrating Koch’s postulates in relation to ChPV and experimentally infected chickens. The enteric diseases caused by bacteria or viruses (RSS) have the particularity of presenting alterations at the ciliated columnar epithelium, Lieberkühn crypts, and in intestinal villi, which show atrophy and fusion or expansion of the lamina propria as a result of inflammatory infiltrates [7,26,33]. Nevertheless, experimental infections with ChPV (ABU-P1) have reproduced RSS, but microscopic alterations were not found in the intestines or in any other organ [22]. In the present investigation, we demonstrated the presence of crypt alterations, which were characterized principally by dilated crypts lined by a squamous epithelium that contained cellular debris and degenerate inflammatory cells, a cyst shape and crypt cell necrosis, all of which are present in the segment of the rolled duodenal loop and in other segments of the intestine; thus, these findings microscopically characterize the ChPV strain isolated in Brazil.

ChPV was detected initially in the intestinal content or in the intestinal wall [6,16,20,34], because the intestine is considered the primary target for enteric virus infection; however, other studies showed the presence of the virus in the spleen and pancreas [35], suggesting another tropism of enteric viruses. These results were supported by qPCR targeting the ChPV strain USP-362-3 [6]. Furthermore, the presence of ChPV was quantified and detected in the three segments of the small intestine (duodenum, jejunum and ileum) and increased after third day post-infection (p.i.) for ileum; after the sixth day for jejunum and after the seventh day for duodenum with high concentrations of genome copies per mg of tissue, however, in the thymus, liver, kidneys, pancreas, proventriculus and bursa, an increase in virus in the tissue was not detected. Nevertheless, the peak of the genomic titer was observed between the first- and fourth-weeks post-infection. The presence of ChPV in many organs until day 42 showed that the virus could persist for a long time, explaining the continuous reinfections through the presence of ChPV in poultry litters [15]. The total weight of the infected animals showed a significant decrease compared to the mock group, which agreed with previously reported data from experimental infection with ABU-P1 [22] and showed that the ChPV strain isolated in Brazil is an important virus causing enteric diseases. Other studies should be performed to understand the interaction of ChPV with the lymphopoietic organs (bursa of Fabricius, spleen and thymus), the exocrine function of digestive organs and certainly the interaction of ChPV with chicken immune system cells. 

## 4. Materials and Methods

### 4.1. Virus Strain, Suspension and DNA Extraction

A strain of ChPV was isolated in SPF chicken embryos as described previously and was designated USP 362–3 [36]. An aliquot of a macerated embryo from which the virus was isolated was transferred to a 2 mL microtube containing 1 mL of 0.1 M PBS at pH 7.4. The suspension was subjected to three freeze thaw cycles consisting of freezing at −80 °C for 10 min and thawing at 56 °C for one min; each cycle was followed by homogenization. The sample was centrifuged at 12,000× *g* for 30 min at 4 °C. The viral DNA was extracted from 250 µL of the supernatant of the viral suspension using TRIzol (Thermo Fisher Scientific, Carlsbad, CA, USA) reagent according to the manufacturer’s instructions. 

### 4.2. Sequencing of the VP1 Gene of ChPV for Molecular Characterization

For molecular characterization of ChPV, PCR was carried out in triplicate as described previously [4], with some modifications. The amplified products were purified using GFXPCR DNA and Gel Band Purification kit (GE Healthcare, Piscataway, NJ, USA) following the manufacturer’s instructions. The purified DNA was then ligated into the PCR™ 2.1-TOPO cloning vector (Thermo Fisher Scientific, Carlsbad, CA, USA), and the vector was transformed into *E. coli* TOP 10 competent cells according to the manufacturer’s instructions. The bacteria were cultured on Luria Bertani (LB) agar containing ampicillin 50 µg/mL. Three colonies were cultured in 3 mL of LB broth and shaken at 230 rpm for 20 h. Plasmid DNA was extracted from the LB broth bacterial suspensions using the QIAprep Spin Miniprep Kit (Qiagen, Hilden, Germany). The plasmid DNA was then sequenced in the forward and reverse directions using a BigDye^®^ Terminator v3.1 Cycle Sequencing kit (Thermo Fisher Scientific, Carlsbad, CA, USA) with M13 primers. Sequencing reactions were performed with an ABI 3730 DNA Analyzer (Thermo Fisher Scientific, Carlsbad, CA, USA). The obtained electropherograms were analyzed using Geneious 11.1.4 software using de novo assembly. The consensus sequences were aligned and compared with other sequences of ChPV available in GenBank using the CLUSTAL W method in ClustalX 2.0.11 software (European Bioinformatics Institute, Saffron Walden, UK). The similarity of nucleotides (nt) and amino acids (aa) was determined using BioEdit 7.1.3. The phylogenetic tree was built using the neighbor-joining statistical method and p-distance substitution model with 1000 bootstraps of replications in the MEGA 7 software package [37].

### 4.3. Experimental Infection

Eighty (*n* = 80) day-old SPF chicks, supplied by CEVA (CEVA Animal Health, Campinas, Brazil), were divided into two groups of forty (*n* = 40) birds: group I was infected with 2 × 10^5^ genome copies (GC) of ChPV in 200 µL of PBS, as previously determined through qPCR [6], and group II was mock infected with 200 µL of sterile 0.1 M PBS pH 7.4. All chicks were challenged by gavage. Both groups were maintained for 42 days in separate isolators under positive air pressure (Alesco, Campinas, SP, Brazil), with water and food supplied ad libitum. The birds were maintained and used in accordance with the guidelines and the approval of the Committee on the Care and Use of Laboratory Animal Resources of the School of Veterinary Medicine, University of São Paulo, Brazil, under protocol number #2569/2012. The birds were observed daily and scored for clinical signs and mortality. The experimental challenge was performed at one day of age, and daily and weekly observations were performed (day 0 pre-inoculation; daily 1, 2, 3, 4, 5, 6 and 7 days post-inoculation; and weekly 14, 21, 28, 35 and 42 days post-inoculation).

### 4.4. Obtaining Organs for Detection of ChPV through qPCR

Five birds (5) at each experimental phase/step were euthanized and subjected to necropsy examination. Hereafter, each organ sample was collected separately, and selected organs included the duodenum, jejunum, ileum, proventriculus, pancreas, liver, spleen, bursa, thymus and kidney. The samples for qPCR were initially stored in liquid nitrogen and then stored at −80 °C until processing. 

### 4.5. Macroscopy and Histopathology Examination

A fragment of each organ listed above from the 7th to 42nd day was fixed in neutral-buffered 10% formalin and embedded in paraffin. Sections of 5 µm thicknesses were prepared and stained with hematoxylin–eosin (H&E). The slides were examined by light microscopy. The intestinal segments were examined by observing the villous:crypt ratio, Lieberkühn crypt morphology, intensity of mononuclear and polymorphonuclear infiltrate in the lamina propria, presence and distribution of lymphoid follicles, number of intraepithelial lymphocytes present in one hundred enterocytes, and the number of cells in mitosis observed in the crypts in three fields at 40×. The microscopic examination of fragments of intestine was semiquantitative according to the parameters described in Table 3. The organs, including the proventriculus, pancreas, spleen, bursa, liver, thymus and kidney, were examined for the presence of inflammatory infiltrate, cell degeneration and other microscopic lesions. 

### 4.6. Detection and Quantification of Chicken Parvovirus in Different Tissues through qPCR

Three randomly selected birds were necropsied from both the infected and mock groups, as described above, and DNA was extracted from 30 mg of tissue from each collected organ. ChPV was detected and quantified using qPCR as described by Nuñez et al. [6]. Each organ was tested in duplicate, and absolute quantification of the ChPV genome copies was performed.

### 4.7. Statistical Analysis

The weight, number of cells in mitosis and number of intraepithelial lymphocytes were evaluated using the Mann–Whitney U test. Semiquantitative evaluations of the variables, namely, the presence of polymorphonuclear and mononuclear cells in the lamina propria and changes in crypt morphology, were evaluated using a chi-square test.

## 5. Conclusions

The genetic characterization of the VP1 gene showed that the Brazilian ChPV isolate belongs to the ChPV II group. Koch’s postulate was demonstrated because RSS was reproduced through detection of the “J” appearance in the duodenum of the infected group, as observed in field conditions.

## Figures and Tables

**Figure 1 pathogens-09-00606-f001:**
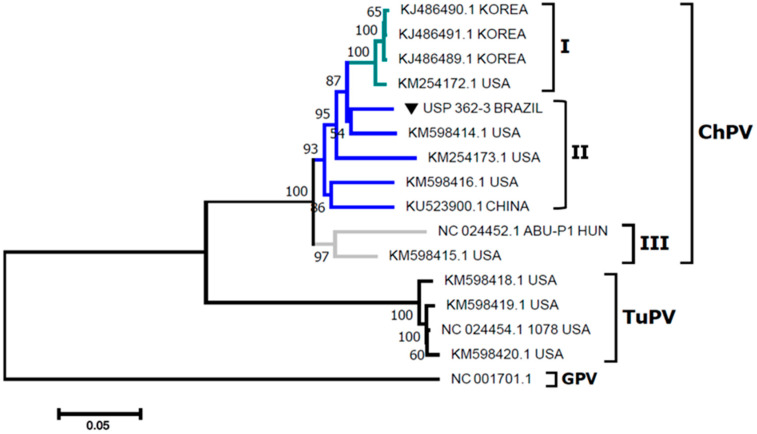
Phylogenetic relationships between the sequences of ChPV obtained in the present work and other sequences of ChPV and turkey parvovirus (TuPV) from Korea, the United States (USA), Hungary (HUN) and China based on complete VP1 gene nucleotide sequences. Sequences were aligned using CLUSTAL W in ClustaX2 2.1. The phylogenetic tree was constructed using MEGA 7 software. The numbers along the branches refer to bootstrap values for 1000 replicates. The scale bar represents the number of substitutions per site. Goose parvovirus (GPV) was used as the outgroup. ▼ = Sequence obtained here.

**Figure 2 pathogens-09-00606-f002:**
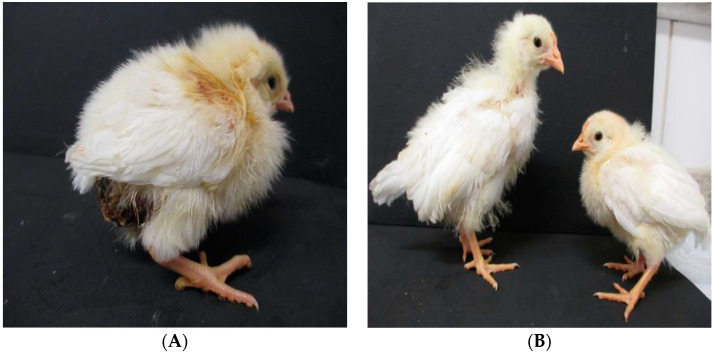
(**A**) Chick from the ChPV-infected group at seven days of age showing clinical signs, such as apathy, somnolence, ruffled feathers, cloacal pasting and diarrhea. (**B**) Birds from the ChPV-infected group showing dwarfism and size disparity.

**Figure 3 pathogens-09-00606-f003:**
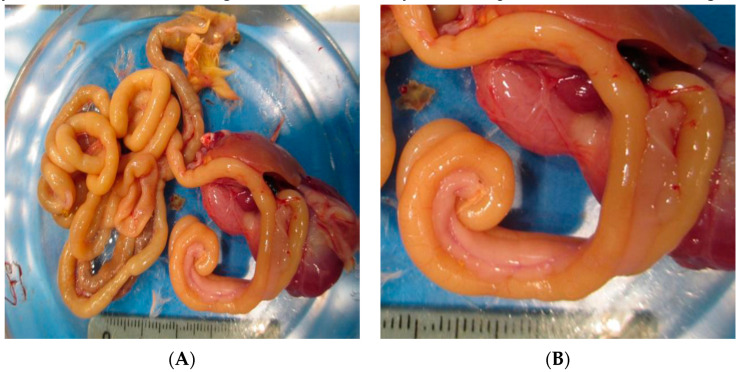
Intestine from ChPV-infected birds at 21 days of age. (**A**) Intestine filled with watery feces, undigested food, and gas and showing intestine segmentation. (**B**) Intestinal loop showing partial volvulus in the mesenteric axis without segment ischemia and resembling the appearance of a corkscrew.

**Figure 4 pathogens-09-00606-f004:**
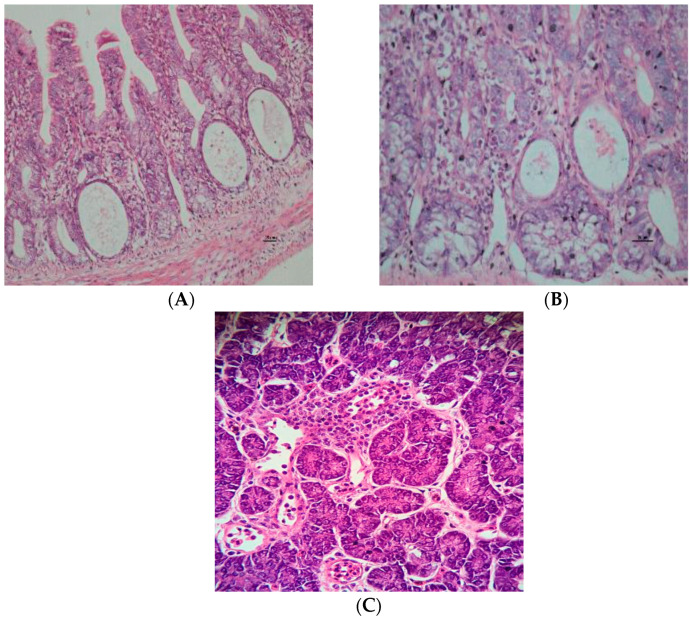
Histopathological analysis of the intestine (jejunum) of infected birds with seven-day-old. (**A**,**B**) Crypts of Lieberkühn lined by a squamous epithelium containing cellular debris-forming structures with the shape of cysts; middle, note the presence of cells in degeneration, the necrosis-surrounding cyst-like structures, and the presence of light infiltration of lymphocytes and plasma cells (enteritis). (**C**) This picture is showing an atrophic pancreas from ChPV-infected birds with infiltrate of lymphocytes and plasma cells; hematoxylin–eosin (HE).

**Figure 5 pathogens-09-00606-f005:**
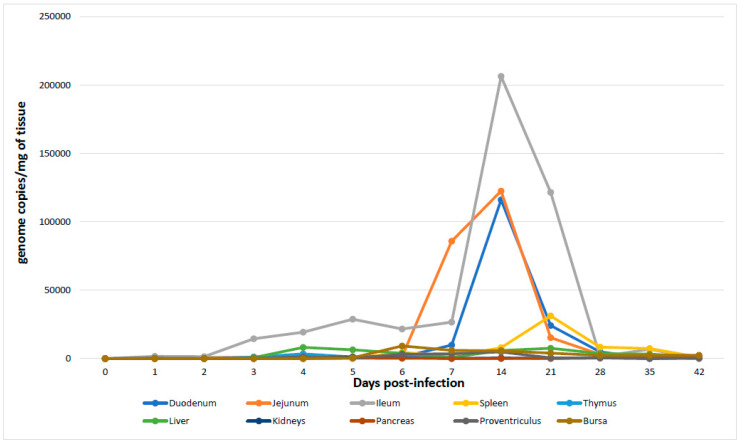
The figure compares the viral distribution in the days after infection with the ChPV strain according to each organ/tissue. Note that peak titer of ChPV is greatest in the intestines. However, it is interesting to note that replication of the virus seems to commence at 2–3 days post-infection for the ileum followed by the jejunum and then the duodenum. Interestingly, multiplication in the pancreas was not observed.

**Table 1 pathogens-09-00606-t001:** Comparison of the nucleotide and amino acid identities of the VP1 sequence of Brazilian isolate of ChPV with other sequences.

	Number	Sequences	Nucleotides (%)		
Numbers of ChPV Sequences		Numbers of TuPV Sequences
1	2	3	4	5	6	7	8	9	10	11		12	13	14	15
ChPV	1	KJ486490.1 Korea	-	99.6	99.8	98.8	95	95	92.3	91.9	92	91.1	88.3		73.5	73.3	73.5	72.9
2	KJ486489.1 Korea	99.8	-	99.8	98.8	95.1	95.1	92.4	92.2	92.2	91.4	88.3		73.6	73.5	73.7	73.1
3	KJ486491.1 Korea	99.8	100	-	98.9	95.1	95	92.3	92.1	92.1	91.2	88.4		73.6	73.4	73.7	73.1
4	KM254172.1 USA	99.8	100	100	-	95	94.8	93	92	91.8	91.7	88.3		73.7	73.5	73.7	73.1
5	USP362-3 Brazil 	98	98.2	98.2	98.2	-	94.9	91.8	91.8	91.8	91.6	88.2		73.6	73.4	73.6	72.9
6	KM598414.1 USA	98.3	98.5	98.5	98.5	97.9	-	91.3	92.7	92.2	91.9	88.7		73.2	73.2	73.4	72.8
7	KM254173.1 USA	96.4	96.5	96.5	96.5	96.2	95.8	-	89.5	89.2	90.7	87.7		73.1	72.9	73.2	72.6
8	KU523900.1 China	96.4	96.5	96.5	96.5	96.7	96.7	94.8	-	91.7	92.4	89		73.1	73.2	73.4	72.9
9	KM598415.1 USA	94.2	94.3	94.3	94.3	94.9	94.9	93.6	95.5	-	90.9	91.9		73.6	73.6	73.8	73.2
10	KM598416.1 USA	96.7	96.8	96.8	96.8	97.7	97	96.2	96.7	95.7	-	88.4		73.6	73.4	73.7	73.1
11	NC_024452.1ABU-P1 HUN	92.8	93	93	93	93	93.3	92.7	94.8	96.8	93	-		73.4	73.1	73.4	72.8
TuPV	12	KM598418.1 USA	79.6	79.7	79.7	79.7	79.7	79.3	78.7	78.7	79.4	79.4	78.8		-	98.3	98.8	98.1
13	KM598419.1 USA	78.8	79	79	79	79	78.5	78.1	78.2	78.8	78.7	78.2		98.2	-	99.5	98.8
14	NC_024454.1 1078 USA	79.6	79.7	79.7	79.7	79.7	79.3	78.8	79	79.6	79.4	79		99.4	98.8	-	99.3
15	KM598420.1 USA	77.8	77.9	77.9	77.9	77.9	77.5	77.1	77.2	77.9	77.6	77.4		97.6	97	98.2	-
**Amino acids (%)**

% = percentage; ChPV = Chicken Parvovirus; TuPV = Turkey Parvovirus; USA = United States; HUN = Hungary; 

 = Sequence obtained here.

**Table 2 pathogens-09-00606-t002:** Clinical signs and macroscopic findings at postmortem examination of chickens from ChPV infected group.

Age of Post-Infection in Days	Clinical Signs								Postmortem Examination													
	Depression	Ruffles Feather	Apathy	Somnolence	Lethargy	Runting	Stunting	Duodenum Loop Enrolled	Foamy Content			Aqueous Feces			Undigested Feed			Intestinal Segmentation			Pancreas Atrophy	Yolk Sac Un-Absorbed
									D	J	I	D	J	I	D	J	I	D	J	I		
0	− (5/5)	− (5/5)	− (5/5)	− (5/5)	- (5/5)	− (5/5)	− (5/5)	− (5/5)	− (5/5)	− (5/5)	− (5/5)	− (5/5)	− (5/5)	− (5/5)	− (5/5)	− (5/5)	− (5/5)	− (5/5)	− (5/5)	− (5/5)	− (5/5)	− (5/5)
1	+ (5/5)	+ (5/5)	+ (5/5)	+ (5/5)	+ (5/5)	− (5/5)	− (5/5)	+ (5/5)	+ (5/5)	+ (5/5)	+ (5/5)	+ (5/5)	+ (5/5)	+ (5/5)	+ (5/5)	+ (5/5)	+ (5/5)	+ (5/5)	+ (5/5)	+ (5/5)	+ (5/5)	+ (5/5)
2	+ (5/5)	+ (5/5)	+ (5/5)	+ (5/5)	+ (5/5)	− (5/5)	− (5/5)	+ (5/5)	+ (5/5)	+ (5/5)	+ (5/5)	+ (5/5)	+ (5/5)	+ (5/5)	+ (5/5)	+ (5/5)	+ (5/5)	+ (5/5)	+ (5/5)	+ (5/5)	+ (5/5)	+ (5/5)
3	+ (5/5)	+ (5/5)	+ (5/5)	+ (5/5)	+ (5/5)	− (5/5)	− (5/5)	+ (5/5)	+ (5/5)	+ (5/5)	+ (5/5)	+ (5/5)	+ (5/5)	+ (5/5)	+ (5/5)	+ (5/5)	+ (5/5)	+ (5/5)	+ (5/5)	+ (5/5)	+ (5/5)	+ (5/5)
4	+ (5/5)	+ (5/5)	+ (5/5)	+ (5/5)	+ (5/5)	+ (4/5)	+ (4/5)	+ (5/5)	+ (5/5)	+ (5/5)	+ (5/5)	+ (5/5)	+ (5/5)	+ (5/5)	+ (5/5)	+ (5/5)	+ (5/5)	+ (5/5)	+ (5/5)	+ (5/5)	+ (5/5)	+ (5/5)
5	+ (5/5)	+ (5/5)	+ (5/5)	+ (5/5)	+ (5/5)	+ (5/5)	+ (5/5)	+ (5/5)	+ (5/5)	+ (5/5)	+ (5/5)	+ (5/5)	+ (5/5)	+ (5/5)	+ (5/5)	+ (5/5)	+ (5/5)	+ (5/5)	+ (5/5)	+ (5/5)	+ (5/5)	+ (5/5)
6	+ (5/5)	+ (5/5)	+ (5/5)	+ (5/5)	+ (5/5)	+ (5/5)	+ (5/5)	+ (5/5)	+ (5/5)	+ (5/5)	+ (5/5)	+ (5/5)	+ (5/5)	+ (5/5)	+ (5/5)	+ (5/5)	+ (5/5)	+ (5/5)	+ (5/5)	+ (5/5)	+ (5/5)	+ (5/5)
7	+ (5/5)	+ (5/5)	+ (5/5)	+ (5/5)	+ (5/5)	+ (5/5)	+ (5/5)	+ (5/5)	+ (5/5)	+ (5/5)	+ (5/5)	+ (5/5)	+ (5/5)	+ (5/5)	+ (5/5)	+ (5/5)	+ (5/5)	+ (5/5)	+ (5/5)	+ (5/5)	+ (5/5)	+ (5/5)
14	+ (5/5)	+ (5/5)	+ (5/5)	+ (5/5)	+ (5/5)	+ (5/5)	+ (5/5)	+ (4/5)	+ (5/5)	+ (5/5)	+ (5/5)	+ (5/5)	+ (5/5)	+ (5/5)	+ (5/5)	+ (5/5)	+ (5/5)	+ (5/5)	+ (5/5)	+ (5/5)	+ (5/5)	+ (5/5)
21	+ (5/5)	+ (5/5)	+ (5/5)	+ (5/5)	+ (5/5)	+ (5/5)	+ (5/5)	+ (4/5)	+ (5/5)	+ (5/5)	+ (5/5)	+ (5/5)	+ (5/5)	+ (5/5)	+ (5/5)	+ (5/5)	+ (5/5)	+ (5/5)	+ (5/5)	+ (5/5)	+ (4/5)	+ (5/5)
28	+ (4/5)	+ (5/5)	+ (5/5)	+ (5/5)	+ (5/5)	+ (4/5)	+ (4/5)	+ (5/5)	+ (5/5)	+ (5/5)	+ (5/5)	+ (5/5)	+ (5/5)	+ (5/5)	+ (5/5)	+ (5/5)	+ (5/5)	+ (5/5)	+ (5/5)	+ (5/5)	+ (4/5)	+ (4/5)
35	+ (5/5)	+ (5/5)	+ (5/5)	+ (5/5)	+ (5/5)	+ (3/5)	+ (3/5)	+ (4/5)	+ (5/5)	+ (5/5)	+ (5/5)	+ (5/5)	+ (5/5)	+ (5/5)	+ (5/5)	+ (5/5)	+ (5/5)	+ (5/5)	+ (5/5)	+ (5/5)	+ (4/5)	+ (5/5)
42	+ (3/5)	+ (5/5)	+ (5/5)	+ (5/5)	+ (5/5)	+ (3/5)	+ (3/5)	+ (5/5)	+ (5/5)	+ (5/5)	+ (5/5)	+ (5/5)	+ (5/5)	+ (5/5)	+ (5/5)	+ (5/5)	+ (5/5)	+ (5/5)	+ (5/5)	+ (5/5)	+ (5/5)	+ (3/5)

+ = positive; − = negative; D = Duodenum; J = Jejunum, I = ileum.

**Table 3 pathogens-09-00606-t003:** Distribution of analyzed histopathological parameters in the fragments of the duodenum, jejunum and ileum from ChPV infected and the negative control.

Organ	Day of Harvest	Group	Mesenteritis	Crypt Morphology						Lamina Propria					
				Crypt Hyperplasia	Cyst-Shape	Necrosis	Number of Cells in Mitosis		*p*-Value	Follicles	Heterophil	Lymphocytes	Intraepithelial Lymphocytes		*p*-Value
**Duodenum**	**7 Days**	**Mock**					6.5	±1.19	* 0.05				4.7	±0.88	* 0.03125
		**ChPV**	+ (5/5)	+ (2/5); ++ (3/5) *	+ (3/5)	+ (5/5)	12	±6.09				+ (1/5)	11.5	1.86	
	14 Days	**Mock**					5.1	±0.92					2.1	±0.37	* 0.03125
		**ChPV**	+ (4/5)	+ (2/5); ++ (3/5) *	+ (3/5)	+ (5/5)	6.4	±2.03				+ (1/5)	11.6	±3.72	
	21 Days	**Mock**					3.4	±0.43	* 0.03125				6.8	±1.3	* 0.02895
		**ChPV**	+ (4/5)	+ (2/5); ++ (3/5) *			11.1	±3.6		+ (2/5)		+ (2/5)	16.7	±1.15	
	28 Days	**Mock**					4.8	±1.12	* 0.03125				6.1	±1.84	
		**ChPV**	+ (5/5)	+ (3/5) *; ++ (2/5)		+ (3/5)	15.9	±4.64		+ (1/5)		+ (3/5) *	15.6	±1.10	
	35 Days	**Mock**					4	±1.22	* 0.03125				6.1	±2.23	
		**ChPV**	+ (5/5)	+ (5/5) *			10.5	±2.06		+ (1/5)		+ (1/5)	13	±2.08	
	42 Days	**Mock**					5.6	±0.79	* 0.02895	+ (1/5)			5.1	±0.69	
		**ChPV**	+ (3/5)	+ (3/5) *; ++ (2/5)	+ (3/5)		17.9	±3.39			+ (1/5)	+ (1/5)	14	±1.25	
**Jejunum**	7 Days	**Mock**					4.4	±1.26	* 0.03125				5.9	±0.88	
		**ChPV**		+ (4/5) *; ++ (1/5)	+ (2/5)	+ (4/5)	9.3	±4.38					7.9	±4.43	
	14 Days	**Mock**					5.5	±1.02					3.7	±1.09	* 0.03125
		**ChPV**		+ (1/5); ++ (4/5) *	+ (3/5)		8.2	±2.63				+ (3/5) *	10.3	±3.21	
	21 Days	**Mock**				+ (5/5)	4.9	±1.12	* 0.03125				10.6	±0.69	* 0.03126
		**ChPV**		+ (5/5) *		+ (5/5)	10.2	±2.93		+ (1/5)		+ (1/5)	16.3	±4.38	
	28 Days	**Mock**					8.1	±1.36					9.9	±2.63	
		**ChPV**		+ (1/5); ++ (4/5) *	+ (1/5)		15.5	±5.69			+ (1/5)	+ (1/5); ++ (1/5)	15	±5.69	
	35 Days	**Mock**					8.3	±1.96					11.1	±3.67	
		**ChPV**	+ (2/5)	+ (3/5) *; ++ (2/5)			10.2	±4.02			+ (1/5)	+ (5/5) *	11.8	±2.38	
	42 Days	**Mock**					6.1	±1.48					4	±0.88	
		**ChPV**		+ (2/5); ++ (2/5)	+ (2/5)		9.5	±2.56				+ (2/5)	11.8	±2.33	
**Ileum**	7 Days	**Mock**					3.6	±0.38	* 0.05				4.9	±1.54	
		**ChPV**		+ (3/5) *; ++ (2/5)	+ (3/5)	+ (4/5)	7.7	±2.81					9.2	±2.96	
	14 Days	**Mock**					4.6	±0.69	* 0.03125				4.9	±0.57	
		**ChPV**		+ (2/5); ++ (3/5) *		+ (2/5)	11.7	±2.92					15.3	±2.96	
	21 Days	**Mock**					4	±1.19	* 0.03125				3.7	±0.33	* 0.04876
		**ChPV**		+ (3/5) *; ++ (2/5)	+ (3/5)	+ (5/5)	7.5	±2.55					11.3	±1.57	
	28 Days	**Mock**					5.1	±0.73					3.9	±1.71	* 0.03125
		**ChPV**		+ (5/5) *		+ (5/5)	12.2	±4.92					10.9	±1.8	
	35 Days	**Mock**					5.7	±0.89	* 0.03125				4	±0.27	* 0.03125
		**ChPV**		+ (3/5) *; ++ (2/5)			13	±3.41				+ (4/5) *; ++ (1/5)	13.8	±0.38	
	42 Days	**Mock**					4.9	±0.51	* 0.03125	+ (2/5)			3.6	±0.5	
		**ChPV**		+ (5/5) *			14.3	±4.86		+ (1/5)		+ (3/5) *; ++ (1/5)	12.1	±2.5	

ChPV = Chicken Parvovirus; * = significant difference; + = mild; ++ = moderate.

**Table 4 pathogens-09-00606-t004:** Detection and quantification of genome copies of ChPV present in the organs harvested during experimental infection.

Tissues/Organs *	Detection and Quantification of ChPV in the Tissues through qPCR at Different Days Post Infection												
	0	1	2	3	4	5	6	7	14	21	28	35	42
Duodenum	−/0	+/31 **	+/81	+/1085	+/1612	+/583	+/1170	+/9863	+/116054	+/24197	+/4921	+/159	+/50
Jejunum	−/0	+/884	+/911	+/665	+/96	+/665	+/1554	+/85808	+/122531	+/15258	+/3100	+/844	+/146
Ileum	−/0	+/1676	+/1468	+/14441	+/19261	+/28822	+/21594	+/26644	+/206642	+/121564	+/1470	+/6855	+/97
Spleen	−/0	+	+	+	+	+	+/4606	+/1570	+/8002	+/31198	+/8364	+/7365	+/1027
Thymus	−/0	+/111	+/50	+/1140	+/3452	+/1267	+/1610	+/56	+/82	+/323	+/688	+/788	+/1767
Liver	−/0	+/54	+/60	+/667	+/8163	+/6430	+/3899	+/832	+/5898	+/7598	+/3684	+/3294	+/1473
Kidney	−/0	+/221	+/33	+/140	+/936	+/1229	+/618	+/101	+/521	+/91	+/473	+/44	+/667
Pancreas	−/0	+/59	+/50	+/62	+/87	+/257	+/210	+/70	+/105	+/230	+/371	+/1078	+/1250
Proventriculus	−/0	+/11	+/21	+/65	+/1374	+/426	+/3124	+/3527	+/4850	+/517	+/489	+/241	+/295
Bursa	−/0	+	+	+	+	+/592	+/9230	+/6000	+/5590	+/3979	+/2345	+/2300	+/2483

* = The results were expressed according to number of genome copies by milligrams of each tissue/organ (genome copies/mg of tissue); ** these numbers represent number of viral genome copies for one mg of tissue. ChPV = Chicken Parvovirus. + = Positive result; − = Negative result.

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
