# Peer review of "Molecular Characterization and Pathogenicity of Chicken Parvovirus (ChPV) in Specific Pathogen-Free Chicks Infected Experimentally"

_pathogens, 2020, doi:10.3390/pathogens9080606_

Round 1

Reviewer 1 Report

This is indeed an interesting, well performed study describing the pathological features of ChPV infections at early stage of age. In fact, disease progression has been monitored over an extended period of time and the authors described the pathological changes in vast detail macroscopically as well as histologically. In addition, they monitored virus loads in different organs over time.

However, there are a few details, this reviewer would like to raise in order to increase the visibility of the observed changes and to potentially allow comparisons with previously published ChPV strains:

  • In agreement with the authors, comparisons of the VP1/2 regions between the different strains are indeed of major interest, since this region harbors information for tissue specificity. Therefore, besides providing information about the overall phylogenetic differences between the different strains, it would be of interest to list the amino acid differences of the different strains in an additional table.
  • Table 2 comprises a lot of features and it is indeed rather difficult to compare “+” and “-“. Here it would help to color code onset and persistence of the diseased feature.
  • Figure 4: To increase the visibility of this figure it would be helpful to highlight the different pathological features with annotated arrows corresponding to the indicated pathology in the figure legend.

Author Response

Dear reviewer, please see attached answer to your comments and/or suggestions. Thanks for your efforts in revised this manuscript.

Reviewer 2 Report

Manuscript Number: pathogens-839825

Title: Molecular Characterization and Pathogenicity of Chicken Parvovirus (ChPV) in Specific Pathogen-Free Chicks Infected Experimentally.

In this work, the authors aimed mainly at reproducing a chicken parvovirus-induced RRS in day-old chicks. They used a field virus isolate that was associated with RRS in chicken flocks. The authors were able to successfully reproduce the syndrome in SPF chicks and -through a 42-day period- described the clinical signs; and macroscopic and microscopic lesions. The authors detected and quantified virus genome copies in numerous body organs over the experimental period. They also reported the full nucleotide sequence of VP1 gene of the ChPV isolate, comparing it to other isolates available on GenBank database. Overall the work is good; however, there are some comments that need to be addressed in the text, which are listed below.

  • Line 15 - Abstract:

The abstract is missing information about the qPCR results in several tissues - please modify accordingly so that these results are briefly represented. If a word limit needs to be maintained, the detailed time points in Line 22 can be removed.

  • Line 24-25: The sentence "The birds were fed with feed and water ad libitum." is not necessary, I would remove it.
  • Line 29: Change "was" to "were".
  • Line 74: The abbreviations NT and AA can be used here as they were mentioned in Line 73; remove the complete words.
  • Line 93 - Table 1:
    • This Table needs to be redone and made simple. There is an overuse of sequence data from the NCBI GenBank database, which is very distracting.
    • I highly recommend that the author only do the NT and AA comparison between the strain they isolated in this study (MK440128.1) and the 15 strains listed in the Table. No need to compare among these 15 strains.
    • Make a simple Table showing the % similarities in NT and AA.
    • It is not clear where the AA% results are located in this Table, though.
    • Line 95: It was mentioned "=Sequence obtained here.", but no symbol was found (before = sign) to indicate this sequence.
    • This Table in its current format with all these results may be used in a review article, instead.
  • Line 97 - Fig 1: Change "relations" to "relationships".
  • Line 103: It was mentioned "=Sequence obtained here.", but no symbol was found (before = sign) to indicate this sequence.
  • Line 122: "...group showing dwarfish animals and with size disparity" may be changed to "...group showing dwarfism and size disparity."
  • Line 131 - Table 2: Some column headings in the Table are written in bold font and others in regular font. Please fix and make them consistent.
  • Line 144 - Figure 3.
    • Please add a picture showing the intestines in the mock group for comparison purposes.
    • Instead of using Left and Right, use A and B to refer to the specific panel.
  • Line 154, after ".....jejunum and ileum" refer to Figure 4, so that the reader can readily find the results.
  • Line 160-161: Move the sentence starting with "Table 3" to line 154 before explaining the results, so that the readers know where the data is located before reading.
  • Line 166-167: The author mentioned that "The birds of the mock group showed segments of the duodenum, jejunum and ileum within the standard of normal histology"
  • However, no pictures are presented to show this. Please add representative pictures to Figure 4.
  • Line 169 - Figure 4:
  • Indicate which part(s) of the small intestine is analyzed here and at what age/day post-infection.
  • Instead of using Left, middle, & Right; use A, B, & C.
  • Add different arrows to each panels to indicate lymphocytes and plasma cell referred to.
  • Add pictures of comparable histological sections from healthy/uninfected birds for comparison.
  • Line 174, no need to mention "hematoxylin-eosin (HE)" as it's not used anywhere within the Figure.
  • Line 186 - Table 3:
  • Please change “C-” under column Group to “Mock”.
  • Change the column heading "Crypt Morphological" to "Crypt Morphology".
  • Change "heterophile" to "heterophil"
  • Some number in the columns "Number of cells in Mitosis" and "Intraepithelial Lymphocytes" are written with comma like 12,0 and 4,0; and some are written with dot like 6.5 and 5.1. Please correct the format or explain why different formats are used.
  • It's not indicated or explained what the sign "++" means. See a relevant comment in #6.
  • Overall, the results need to be modified for easier understating and also the results of the mock (C-) group are not reported clearly in the Table. There is no need to use the + sign. Just using 3/5 or 5/5 will clearly indicate that 3 out of 5 or 5 out of 5 birds were positive for that parameters. Also, please report all results even if the value is 0/5.
  • Line 188, change "significative" to "significant".
  • Line 190:
  • I recommend modifying the title to "Detection and Quantification of Chicken Parvovirus genome copies in the Different Tissues"
  • Please avoid using viral particle in the manuscript as the qPCR assay cannot be used to quantify virus particles, unless implemented in a functional assay that can link genome copies to virus infectivity.
  • The author is referred to this article https://www.sciencedirect.com/science/article/pii/S0166093416303500, for more info about the topic.
  • Line 192: Use "Peak titers of ChPV", instead of "Peak shedding". Please avoid using virus shedding in the manuscript as it is not relevant to the samples collected. Shedding is not suitable here because viruses are only being shed from body vents, i.e. nasal/oral and cloacal.
  • Line 194: I recommend changing "multiplication of viral particles" to "replication of the virus" as a more standard virology expression.
  • Line 198: Change "viral particles of ChPV" to "genomic titer of ChPV".
  • Line 199: Change "virus particle" to "virus genomic titer".
  • Line 203 - Table 4:
  • Please modify "quantification of viral particles" to "quantification of viral genome copies"
  • Modify the middle row title by adding "....through qPCR at different days postinfection".
  • Line 207, change "viral particles" to "viral genome copies"
  • Line 211 – Figure 5:
  • Please provide a better quality of the Figure. The information on the figure are very difficult to read.
  • Line 212: Use "peak titer" instead of "peak shedding".
  • Line 213: I recommend changing "multiplication of viral particles" to "replication of the virus" as a more standard virology expression.
  • Line 268-269: Change "...peak shedding of viral particles was observed..." to "...peak genomic titer of ChPV was observed..."
  • Line 292: “ coli should be italicized.
  • Line 312, "ad libitum" should be italicized.
  • Line 340: As mentioned earlier, please change "ChPV viral particles" to "ChPV genome copies".

END

Author Response

(The authors gave the same response as above.)
